# Developing New Agents for Treatment of Childhood Cancer: Challenges and Opportunities for Preclinical Testing

**DOI:** 10.3390/jcm10071504

**Published:** 2021-04-04

**Authors:** Samson Ghilu, Raushan T. Kurmasheva, Peter J. Houghton

**Affiliations:** Greehey Children’s Cancer Research Institute, 8403, San Antonio, TX 78229, USA; ghilu@uthscsa.edu (S.G.); kurmasheva@uthscsa.edu (R.T.K.)

**Keywords:** Patient-derived xenografts, pediatric cancer, drug development, single mouse design

## Abstract

Developing new therapeutics for the treatment of childhood cancer has challenges not usually associated with adult malignancies. Firstly, childhood cancer is rare, with approximately 12,500 new diagnoses annually in the U.S. in children 18 years or younger. With current multimodality treatments, the 5-year event-free survival exceeds 80%, and 70% of patients achieve long-term “cure”, hence the overall number of patients eligible for experimental drugs is small. Childhood cancer comprises many disease entities, the most frequent being acute lymphoblastic leukemias (25% of cancers) and brain tumors (21%), and each of these comprises multiple molecular subtypes. Hence, the numbers of diagnoses even for the more frequently occurring cancers of childhood are small, and undertaking clinical trials remains a significant challenge. Consequently, development of preclinical models that accurately represent each molecular entity can be valuable in identifying those agents or combinations that warrant clinical evaluation. Further, new regulations under the Research to Accelerate Cures and Equity for Children Act (RACE For Children Act) will change the way in which drugs are developed. Here, we will consider some of the limitations of preclinical models and consider approaches that may improve their ability to translate therapy to clinical trial more accurately.

## 1. Introduction

Approximately 70% of children with childhood can be cured using multimodality treatments, but the outcome is still poor for those with advanced or metastatic disease [1,2,3]. Specifically, whereas the outcome for patients with localized disease is good, the 5-year event-free survival rates are 30 percent or less in children with metastatic or relapsed Ewing sarcoma, osteosarcoma, or rhabdomyosarcoma, and intensive chemo-radiotherapy has not substantially altered this outcome over the past three decades [2,4,5,6]. Similarly, outcome for infants with malignant rhabdoid tumors (MRT), advanced brain tumors, including glioblastoma and diffuse intrinsic pontine glioma (DIPG) remain dismal [7,8]. As additional cytotoxic drugs alone are unlikely to significantly increase cure rates, alternative and complimentary approaches need to be explored. Further, given the long-term toxicities associated with intensive dose chemotherapy/radiation in the pediatric population, new therapies associated with less genotoxicity are needed [9,10,11,12].

The failure of current therapeutic approaches to effectively treat advanced disease is exemplified by 30 years of clinical experience managing metastatic rhabdomyosarcoma where survival rates have not increased meaningfully despite numerous cooperative group trials (IRS studies I-IV). Only 30% of patients with stage IV disease experience long-term survival, even with the use of intensive multimodality treatment. This is further emphasized by the rapid failure of patients enrolled on the most recent high-risk rhabdomyosarcoma study (ARST0431), where patients diagnosed with metastatic disease, failed during the 52 weeks of treatment (25%) or had an event within two years (80%) [2]. Similarly, the prognosis for patients with advanced osteosarcoma and Ewing sarcoma remains poor; patients develop resistance to current therapies, often associated with systemic toxicity [13]. These results argue that we have to develop innovative approaches to identify novel targets. Identification of new targets requires a greater understanding of the underlying biology of these sarcomas, and a strategy for developing novel agents in appropriate preclinical models. 

The predominant preclinical models used for drug development are cells in culture and human tumor xenografts grown in immunodeficient mice. In early days of the NCI screening program, cell-line derived xenografts were used, with a single line representing a tumor type (MX-1 represented breast cancer, for example). Clearly, with hindsight where genomic analysis has revealed the complexity of these diseases, such an approach seems naïve, although regarded as state-of-art at that time. Further, cells in culture seem to deviate rapidly from the original cancer, hence may no longer represent the original tumor. Patient-derived xenografts (PDX’s) wherein a patient tumor is transplanted directly into an immune deficient rodent and used for testing on early passages have become the predominant preclinical models for drug development. The relative ‘ease’ for establishing PDX models allows preclinical testing to encompass greater genetic/epigenetic diversity into preclinical trials [14]. However, these models are resource-constrained (cost) and often testing is slow due to the growth rate of the model.

However, all models have deficiencies that limit their value in translating preclinical results to successful clinical application. For PDX models in mice that lack a competent immune system, there are limitations on their use for developing immuno-oncology agents, although ‘humanized’ mice are being developed. Here, we will discuss some of the limitations of PDX models in development of conventional drugs and biologics that target tumor cells, rather than enhance immune reactions against the cancer, and the potential to develop approaches to overcome these deficiencies allowing more accurate translation of preclinical results. 

### 1.1. Differences in Species Tolerance

In most cases rodents are more tolerant of drugs than are humans. Hence, at the maximum tolerated dose of a drug in mice, human tumor xenografts are exposed to drug levels that cannot be achieved in patients, hence the models overpredict efficacy [15]. Consequently, it is important to simulate clinical exposures for drugs in preclinical testing. Usually, testing in pediatric preclinical studies lags behind adult clinical trials, and for some drugs results of adult phase I pharmacokinetic studies are available to guide both dose and schedule of administration. Alternatively, approaches such as allometric scaling in several species can be valuable to predict patient dosing. For some drugs pharmacokinetics between mouse and patients is very similar at the respective species maximum tolerated dose (MTD), for example melphalan [16]. On the other hand, several histone deacetylase inhibitors have different pharmacology in mice and humans and are difficult to model accurately in preclinical testing [17]. An example is shown in Figure 1, where the plasma exposure to acylfulvene, an eludin S derivative developed some years ago, is shown for dose escalation to its MTD in patients [18], the exposure at its MTD in mice, and drug exposure at a fraction of the MTD where the drug failed to elicit tumor regression in any of the nine brain tumor models examined [19]. The importance of relating drug exposure to accurate translation of preclinical results was demonstrated subsequently for topotecan in the treatment of children with neuroblastoma [20]. Alisertib, an Aurora kinase A inhibitor, serves as a further example where systemic exposures in mice provide the basis for failure to accurately translate preclinical results. Alisertib (MLN8237) showed broad spectrum activity at its MTD when tested by the Pediatric Preclinical Testing Program (PPTP) [21] where systemic exposures to alisertib were ~8-fold greater in mice than in patients, and where the drug had a limited range of efficacy ~3-fold from the MTD [22,23]. Of note, retesting alisertib at doses and schedules that gave systemic exposures in mice similar to that in the clinical trial, showed little activity against ALL models [23]. Using the relative exposures at the respective MTD in mice and human, and the effective dose range for the agent in mice, seems to be a metric that distinguishes drugs that are active in preclinical models but fail in the clinic, compared to those where preclinical activity translates more accurately, Table 1. From PPTP experience, if a drug causes objective tumor regressions only at the mouse MTD, it is unlikely to have clinical activity [24,25]. These examples stress the need for evaluating novel agents in mice at doses and schedules of administration that simulate clinical drug exposures. For many drugs tested through the PPTP and PPTC (Pediatric Preclinical Testing Consortium) programs adult phase I pharmacokinetic data were available. Where human pharmacokinetic data are not available, systemic exposures in mice should at least be compared to another larger species (e.g., dog) to gauge whether exposures in mice at doses causing tumor shrinkage are relevant. 

### 1.2. Preclinical Models Do Not Accurately Recapitulate Clinical Disease

Genetic studies have demonstrated that cancers once considered homogeneous entities are in fact comprised of genetic subtypes, some of which have drug vulnerabilities. For many years it has been recognized that subgroups exist for neuroblastoma (MYCN amplified), or rhabdomyosarcoma (fusion-positive and fusion-negative), however with the advent of genome sequencing and expression profiling, it is now possible to subtype many other cancers including brain tumors such as medulloblastoma, or ependymoma where four and nine molecular subgroups have been proposed, respectively [26,27]. While such genetic subtyping may identify groups with specific drug vulnerabilities (e.g., Smoothen inhibitors in PCTH mutated medulloblastoma [28]) it further complicates clinical trials, as each subgroup represents only a fraction of an already small number of patients. It also increases the requirement for development of preclinical models that encompass the specific genetic subtype. 

Development and molecular characterization of large numbers of pediatric cancer PDX models has been undertaken both in Europe and the US by the Innovative Therapies for Children with Cancer project–Pediatric Preclinical Proof of Concept Platform (ITCC-P4), and the PPTP/C. The PPTP/C has established and characterized over 700 pediatric cancer PDX models. Similarly, the Childhood Solid Tumor Network [29] has developed and characterized many new models. Ongoing studies at the Greehey Children’s cancer Research Institute (GCCRI) have establish a further 140 PDX models (approximately equal numbers of leukemias and solid tumors) focused on cancers from underserved Latino populations in south Texas. Thus, large numbers of well-characterized pediatric PDX models are available for the community, although the number of models available for study of extremely rare pediatric cancers remains problematic. 

An ongoing initiative is to build a web-based inventory of all adequately characterized pediatric and adult PDX models is ongoing (PDXNet; https://www.pdxnetwork.org, accessed on 29 March 2021). Support for the use of PDX preclinical models is based upon early studies where models identified known clinically active agents and prospectively identified novel drugs and combinations that subsequently demonstrated clinical utility [16,30,31,32,33]. Studies by Lock and colleagues [34,35,36] established the value of acute lymphoblastic leukemia (ALL) PDX models, demonstrating the genetic fidelity and value for developmental therapeutics [37] and identifying drug resistance mechanisms [38,39]. Results from these PDX models led to the National Cancer Institute-sponsored PPTP, which evaluated over 80 drugs or drug combinations in up to 50 pediatric cancer xenograft models [40]. Summary results from the PPTP [37,41,42] demonstrate some principles that may relate to how new agents are developed in pediatric cancer.

The major limitation of the PPTP studies is that while 50 in vivo models were used to represent ‘childhood cancer’ relatively few models were used to represent the genetic diversity of any one entity. For example, six models were used to represent ‘kidney cancer’. Three were Wilms tumors, and three were malignant rhabdoid tumors. For ALL, eight models were used and for neuroblastoma six models. Clearly these numbers cannot simulate the clinical heterogeneity of any cancer type. 

### 1.3. Encompassing Genetic Heterogeneity of Clinical Cancer

The challenge is to address encompassing adequate genetic/epigenetic diversity into preclinical drug testing. Indeed, under the Research to Accelerate Cures and Equity for Children Act (RACE For Children Act), FDA may mandate clinical assessment if the target of a drug being developed is substantially relevant to the growth and progression of childhood cancer. However, response to a targeted drug may be context-specific. For example, the response of BRAF^V600E^ melanoma compared to colon cancer with the same driver mutation [43,44], or the objective response rate of pediatric BRAF-mutant low-grade glioma [45] compared to glioblastoma with BRAF^V600E^ mutations [46]. Thus, there will be a requirement for testing in multiple models representing the same genetic variant.

One approach to encompassing genetic diversity for a given cancer type is to use a single mouse testing (SMT) design, rather than conventional experimental designs that use 8–10 mice for control and treatment groups [40,47]. Validity of the SMT design is based upon a retrospective analysis of 2134 tumor-drug studies undertaken by the PPTP, where the response of a tumor in one mouse, selected at random from the group, was compared to the median group response. This analysis showed that the SMT accurately predicted in 78% of studies. Allowing for a deviation of ± one response classification (e.g., Stable Disease [SD)] versus Partial Response [PR]), the concordance was 95%, Figure 2 (left). Further, the SMT analysis was accurate in identifying the antitumor activity of 66 of 67 drugs in terms of the Objective Response Rate (ORR) determined for each drug over a range of tumor models, Figure 2 (right). Prospective studies with up to 90 ALL models and up to 50 solid tumor models have shown that SMT has similar concordance with conventional testing. Importantly, using SMT one can potentially incorporate up to 20-fold the number of models for evaluation of an agent, encompassing many diseases, or encompassing the genetic diversity of a given disease. Examples of the former design is shown in Figure 3, where the antitumor activity of an antibody conjugate (trastuzumab-deruxtecan; DS-8201a) was evaluated in 35 xenograft models. Tumor sensitivity to DS-8201a varied considerably with some tumors progressing on treatment or having short event-free survival (EFS) as shown by the Kaplan–Meier analysis. As each tumor model is linked to whole exome sequencing (WES) and RNA sequencing (RNAseq), such results can be of value for identification of biomarkers associated with drug sensitivity or resistance. Of note, five models that remained in Complete Response at the end of the observation period (20 weeks), were malignant rhabdoid tumors, perhaps indicating sensitivity of this disease to DS-8201a [48]. 

While the study above shows the activity of an agent across tumor types, an alternative approach is to test an agent in a large number of models representing the genetic diversity of a specific tumor type. This was an approach first reported by the group at Novartis [49] using some 30 melanoma PDX models. An example of evaluating combination chemotherapy (vincristine/actinomycin D/cyclophosphamide; VAC), the ‘gold standard’ for treatment of rhabdomyosarcoma, against 34 models of this disease is shown in Figure 4. This is an ongoing experiment, but clearly shows that some models are intrinsically resistant to treatment, whereas others are highly sensitive. The rhabdomyosarcoma models encompass fusion-positive (alveolar), fusion-negative (embryonal), with and without RAS mutations. Importantly, these types of experiments may be valuable in identifying biomarkers that segregate with resistance. Perhaps of equal importance, is that the experiments shown used 35 and 34 mice, respectively whereas to gain the same results using conventional testing would have used over 600 mice per study. Similarly, correlating responses of Ewing sarcoma models to novel agents using SMT with characteristic STAG2, CDKN2A, and TP53 mutations [50] could provide important prognostic insights. 

In SMT, because there is no control tumor, one has to consider the potential for spontaneous tumor regression of a xenograft potentially biasing the study. We have randomly analyzed growth of 1147 control tumors from the GCCRI tumor panels and observed only one tumor regression (0.087%). Thus, under the experimental conditions used, spontaneous regression (as opposed to drug-induced regression) is extremely rare. 

### 1.4. Criteria for Defining ‘Antitumor Activity’ Is Less Stringent in Preclinical Studies Than in Clinical Trials

Antitumor activity in clinical trials is based on RECIST criteria and determined by two main metrics, tumor regression and time-to-tumor progression (TTP; [51]). Rarely is a placebo group alone incorporated into trials, rather the outcome for two therapies (experimental vs. standard of care), or two experimental arms will be compared (phase II of temsirolimus vinorelbine cyclophosphamide versus bevacizumab vinorelbine, cyclophosphamide; [52]) or a historical control is used.

For preclinical studies, there are no standardized criteria for evaluating antitumor activity. The PPTP developed a set of criteria so that drugs could be evaluated in a standard manner, allowing at least some comparisons to be made between agents. However, there are no criteria that are uniform between journals that specify what constitutes antitumor activity of an agent. It is thus not uncommon to see marginal antitumor responses, say 20% inhibition of tumor growth, being presented as validation of a molecular target. Alternatively, tumor growth delay over a short period (say 7–8 days) is presented, which is irrelevant to accurate clinical translation, and may not even cover the period where the drug may induce toxicity. For example, bone marrow toxicity in mice is manifest about day 15 for many cytotoxic agents.

The other issue is how data are presented in publications. In most cases, primary data are not presented, rather ‘manipulated’ such that tumor volume (mean ± SE) or treated tumor volumes at a set time point relative to control volumes are presented. While a result showing 50% decrease in treated tumor volume relative to the control group may look impressive, it would represent progressive disease if such a level of antitumor activity were observed in a clinical trial. Progressive disease is defined as 25% increase in tumor dimensions in clinical trials. That a drug causes a difference in EFS compared to the control EFS at a *p* = 0.05 level of statistical significance is biologically meaningless. To cite the American Statistical Association (ASA) “No single index should substitute for scientific reasoning” [53]. There should be standardized metrics for assessing antitumor activity in preclinical studies based on biological significance of the observed response, as for clinical studies [54].

Similar to policies of journals, there are no antitumor metrics provided by clinical trials groups, such as the Children’s Oncology Group (COG), to guide preclinical drug development and defining criteria required for advancing an agent to clinical evaluation. Do we accept antitumor activity in one tumor model, or should there be robust activity (tumor regression, prolonged EFS, etc.) in multiple models? For future studies, criteria for antitumor activity, together with evidence that drug doses used in mice are relevant to exposure in humans need to be developed. While one could argue that such studies with multiple tumor models may be expensive, and maybe beyond the capabilities of many academic laboratories, it is important to consider these additional preclinical costs relative to costs involved in developing a clinical trial, and the costs of wasted patient resources for a trial that does not demonstrate efficacy [55,56]. On a positive note, harmonizing criteria for preclinical testing between the U.S. (PPTC) and European (ITCC-P4) programs is ongoing. 

### 1.5. Unpredicted Toxicities in Patients

It is estimated that 95% of cancer drugs under development fail during the course of clinical development. For pediatric clinical trials, the majority fail because of lack of efficacy rather than toxicities. In part this is a consequence of phase I trials in adults preceding those in children, where unexpected or unacceptable toxicity may be identified, and development terminated prior to pediatric evaluation. However, recent development of MDM2 inhibitors serves to illustrate that how a drug is developed may define and limit the value of preclinical models in the development process. In developing MDM2 inhibitors, human MDM2 and TP53 were used to screen small molecules that prevented interaction of these proteins and ubiquitylation and degradation of TP53. However, affinity for human MDM2 greatly exceeded that for murine MDM2 [57]. These inhibitors demonstrated minimal toxicity in non-human species but caused severe thrombocytopenia in patients [58]. Similarly, inhibitors of the antiapoptotic protein MCL1 have greater affinity for human MCL1 compared to murine protein, thus making it difficult to assess therapeutic index in the xenograft models without engineering the mouse to express only human MCL1. Consequently, with drugs developed specifically against human proteins, the lack of toxicity in preclinical models may be misleading.

## 2. Summary

Patient derived xenograft models have utility in identifying drugs and drug combinations if used correctly. Major failures in accurately translating data from these models to clinical practice appear largely a consequence of poor experimental design, failure to appreciate potential differences in pharmacology, and a failure to use clinically meaningful criteria for assessing the antitumor activity of agents. For PDX models where growth characteristics are consistent, we propose the single mouse testing (SMT) approach that overcomes some of the limitations of conventional testing. SMT is valuable when large antitumor effects are sought—effects that may more readily translate into clinical activity. Importantly, SMT allows for the screening of a large number of tumor models representing diverse tumor types, or a similar number of models representing genetic/epigenetic heterogeneity of a specific childhood cancer, within the resource constraints currently available for conventional testing of relatively few models.

## Figures and Tables

**Figure 1 jcm-10-01504-f001:**
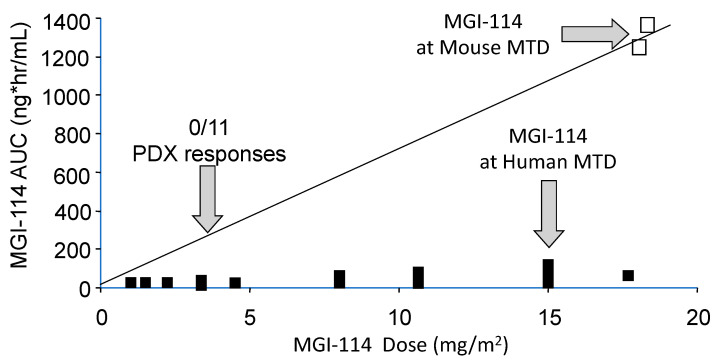
Relative systemic exposures to MGI-114 (acylfulvene) in mice and human. Human and mouse pharmacokinetic data are from Eckhardt et al., 2000 [18], and Leggas et al., 2002 [19], respectively. The AUC (estimated) at the dose of MGI-114 where all 11 PDX models progressed on treatment is shown by the left vertical arrow.

**Figure 2 jcm-10-01504-f002:**
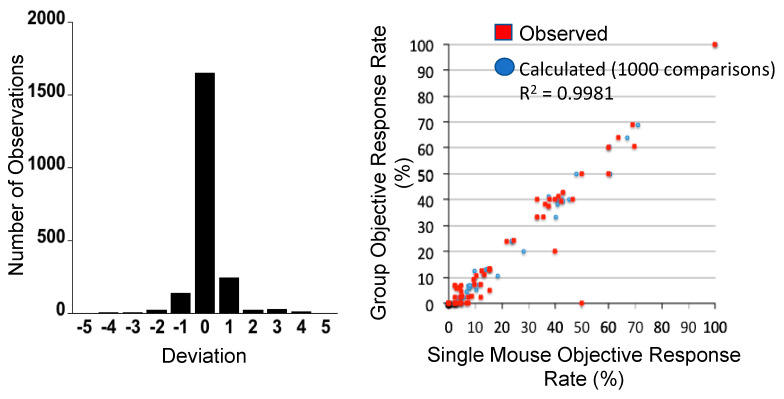
**Left***:* Distribution of deviation for 2134 observations. Single mouse prediction of response was compared to the median response for groups of tumor-bearing mice. A score equaling zero indicates accurate prediction, whereas +1 and −1 refer to over- and under-prediction by one response classification. **Right**: Objective response rates (ORR) were calculated for all tumor models tested for a particular drug (1 to 55 models) for different studies, based upon the Group Median response; (red) responses predicted from a randomly chosen single mouse are plotted against group median response; (blue) the Single Mouse ORR mean ORR correlation based on 1000 single mouse samples. (From [48] with permission).

**Figure 3 jcm-10-01504-f003:**
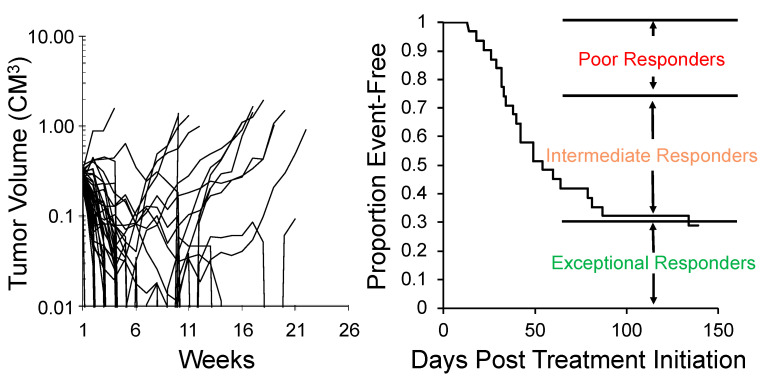
SMT evaluation of trastuzumab-deruxtecan (DS-8201a). **Left**: Tumor volume response for 35 xenograft models treated with DS-8201a. Each curve represents the growth of an individual tumor. **Right**: Time to event is replotted as a Kaplan–Meier EFS Probability curve, allowing classification of tumors as poor-, intermediate- or exceptional-responders. Note 5 malignant rhabdoid tumor (MRT) models were maintained CR at week 25.

**Figure 4 jcm-10-01504-f004:**
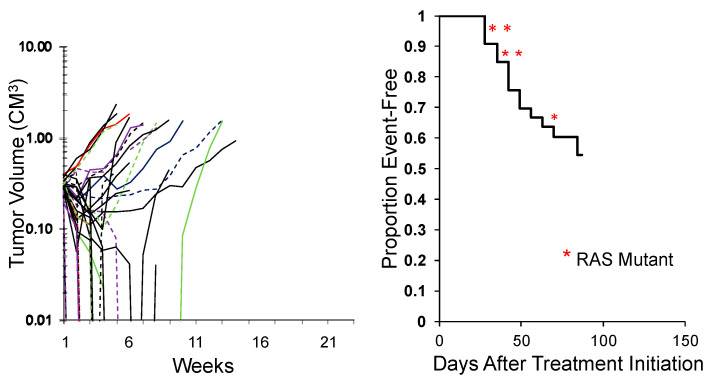
**Left**: Initial responses of 34 RMS models to Cycle 1 vincristine + actinomycin D + cyclophosphamide (VAC) using the SMT experimental design (ongoing expt.). **Right**: Kaplan–Meier plot for EFS. Red asterisks: RAS mutant ERMS models.

**Table 1 jcm-10-01504-t001:** Relative drug exposures at the mouse and human MTD, and efficacy range in mice.

Agent	AUC @ Mouse MTD AUC @ Human MTD	Effective Dose Range from MTD in Mouse
Clinically Not Active		
DMP-840	15–20	~2–3
Carzelesin	~80	<2
Sulophenur	~8	~3
MGI-114	>15	~3
Alisertib (MLN8237)	~8	~3
Clinically Active		
Melphalan	1	~2–3
Topotecan	~3	~10
Irinotecan	16	~100

## Data Availability

Not Applicable.

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
