# Peer review of "Developing New Agents for Treatment of Childhood Cancer: Challenges and Opportunities for Preclinical Testing"

_jcm, 2021, doi:10.3390/jcm10071504_

Round 1

Reviewer 1 Report

So this is very much a review article of the current status of preclinical testing for childhood cancer. It is not a scientific paper in its own right with new data and hypotheses. It is a review of the subject and I think it is well balanced and comprehensive in its review. The paper lists the various pros and cons of a range of preclinical testing models used to date. As a review article it is interesting and a useful amalgamation of current knowledge in the field It is a very well written article. For many people it will be a useful educational article. I am not aware of similar review articles on the subject.

This is an excellent review article and I have no additional comments for the authors.

Author Response

No edits were suggested.

Reviewer 2 Report

I would like to congratulate the authors on a well-written manuscript on the challenges faced in pediatric oncology to bring new drugs to the market and the role better preclinical testing can have in advancing opportunities for children with challenging or relapsing disease.

I have two suggestions and minor technical edits.

  1. The section on SMT design should have its own heading and reflect it is presented as one of the solutions for how to move forward. It was not until the summary that I realized that SMT was the solution being proposed by the authors. I think the several paragraphs and figures on SMT should be taken out from the “1.2 preclinical models do not accurately recapitulate clinical disease” and put as a separate section.
  2. There are two moments when authors should revise language to avoid sounding condescending. In line 53, sentence reads “seems naïve”. There should be an explanation for ow/why this is and/or a reference for the statement. In line 287, sentence includes “used intelligently”. This should be removed or rephrased.

Minor suggestions:

  1. Line 10 = change to “… and 70% of patients achieve long-term “cure”, ….” (instead of are long-term “cures”; perhaps this used to read “are long-term survivors”)
  2. Line 45-46 = add approaches to the “…we have to identify novel targets”, since we also need novel approaches to find novel targets.
  3. Lines 49-57 – this paragraph needs some revising. It is a mix of statements and it does not currently flow very well
  4. Lines 61-67 – there are two statements of the aim, both starting with “Here we will…”. These should be collapsed or one of the statements rephrased
  5. Line 232 – change “an historical control” to “a historical control”
  6. Line 266 – change U.S to U.S. (second period is missing)

Author Response

1. The section on SMT design should have its own heading and reflect it is presented as one of the solutions for how to move forward. It was not until the summary that I realized that SMT was the solution being proposed by the authors. I think the several paragraphs and figures on SMT should be taken out from the “1.2 preclinical models do not accurately recapitulate clinical disease” and put as a separate section.

            Response: Good idea! Section 1.3. 1.3. Encompassing genetic heterogeneity of clinical cancer

2. There are two moments when authors should revise language to avoid sounding condescending. In line 53, sentence reads “seems naïve”. There should be an explanation for ow/why this is and/or a reference for the statement. In line 287, sentence includes “used intelligently”. This should be removed or rephrased.

Response: Reworded: 1) Clearly, with hindsight where genomic analysis has revealed the complexity of these diseases, such an approach seems naïve, although regarded as state-of-art at that time. 2) Intelligently replaced with ‘correctly’.

Minor suggestions:

1. Line 10 = change to “… and 70% of patients achieve long-term “cure”, ….” (instead of are long-term “cures”; perhaps this used to read “are long-term survivors”)

Response: Corrected as suggested

2. Line 45-46 = add approaches to the “…we have to identify novel targets”, since we also need novel approaches to find novel targets.

Response: Edited as suggested

3. Lines 49-57 – this paragraph needs some revising. It is a mix of statements and it does not currently flow very well

Response: Section expanded to be clearer.

4. Lines 61-67 – there are two statements of the aim, both starting with “Here we will…”. These should be collapsed or one of the statements rephrased

Response: Two statements collapsed to one.

5. Line 232 – change “an historical control” to “a historical control”

Response: Corrected

6. Line 266 – change U.S to U.S. (second period is missing)

Response: Corrected

Reviewer 3 Report

Ghilu S, et al present a review of the challenges in pre-clinically models used to prioritize novel agents for evaluation in pediatric cancer clinical trials.  The manuscript provides a high-level overview of the process, specific examples to illustrate issues, and recommendation for improvements.  The review is a valuable document of the current state with guidance for the future.  

Line 25: “Advanced” has little meaning in most pediatric sarcomas, since therapy is generally the same for localized tumors regardless of size or regional extension.  It would be more accurate to contrast initially localized sarcomas to metastatic or recurrent tumors.

Line 30: add reference for poor outcome in selected pediatric brain tumors.

Line 41: Oberlin scoring as risk stratification in metastatic RMS is not defined in the manuscript and too obscure for a review article.  Remove and replace with more general discussion of the poor outcome for metastatic RMS overall.

Line 104: Three arrows are in Figure 1; which is “arrowed”?

Line 110: Should the mouse to human AUC for irinotecan be 16-100?

Line 160: add references to support the tumor context specific activity of BRAF inhibitors among tumors despite the same BRAF V600E mutation.

Line 194: have the SMT data with DS-8201a been published?  If so, add reference.

Line 303: Replace reference #2 with a more general reference regarding the outcome for RMS, such as PMID: 30617281.

Line 448: Replace abstract reference with primary manuscript: PMID: 31513481.

Author Response

Line 25: “Advanced” has little meaning in most pediatric sarcomas, since therapy is generally the same for localized tumors regardless of size or regional extension. It would be more accurate to contrast initially localized sarcomas to metastatic or recurrent tumors.

Response: Corrected as requested.

Line 30: add reference for poor outcome in selected pediatric brain tumors.

Response: References added.

Line 41: Oberlin scoring as risk stratification in metastatic RMS is not defined in the manuscript and too obscure for a review article. Remove and replace with more general discussion of the poor outcome for metastatic RMS overall.

Response: Edited as suggested.

Line 104: Three arrows are in Figure 1; which is “arrowed”?

Response: ‘horizontal’ arrow added.

Line 110: Should the mouse to human AUC for irinotecan be 16-100?

Response: Table corrected.

Line 160: add references to support the tumor context specific activity of BRAF inhibitors among tumors despite the same BRAF V600E mutation.

Response: References added.

Line 194: have the SMT data with DS-8201a been published? If so, add reference.

Response: Reference added.

Line 303: Replace reference #2 with a more general reference regarding the outcome for RMS, such as PMID: 30617281.

Response: Reference replaced.

Line 448: Replace abstract reference with primary manuscript: PMID: 31513481.

Response: Reference replaced.